# Odor-evoked transcriptomics of *Aedes aegypti* mosquitoes

**Fredis Mappin, Anthony J. Bellantuono, Babak Ebrahimi, Matthew DeGennaro***

Department of Biological Sciences & Biomolecular Sciences Institute, Florida International University, Miami, Florida, United States of America

* mdegenna@fiu.edu

**Data Availability Statement:** The RNA-seq transcriptome data is available from the NCBI Sequence Read Archive (SRA) and are associated with BioProject ID PRJNA942345. All other

## Abstract

Modulation of odorant receptors mRNA induced by prolonged odor exposure is highly correlated with ligand-receptor interactions in *Drosophila* as well as mammals of the Muridae family. If this response feature is conserved in other organisms, this presents an intriguing initial screening tool when searching for novel receptor-ligand interactions in species with predominantly orphan olfactory receptors. We demonstrate that mRNA modulation in response to 1-octen-3-ol odor exposure occurs in a time- and concentration-dependent manner in *Aedes aegypti* mosquitoes. To investigate gene expression patterns at a global level, we generated an odor-evoked transcriptome associated with 1-octen-3-ol odor exposure. Transcriptomic data revealed that ORs and OBPs were transcriptionally responsive whereas other chemosensory gene families showed little to no differential expression. Alongside chemosensory gene expression changes, transcriptomic analysis found that prolonged exposure to 1-octen-3-ol modulated xenobiotic response genes, primarily members of the cytochrome P450, insect cuticle proteins, and glucuronosyltransferases families. Together, these findings suggest that mRNA transcriptional modulation of olfactory receptors caused by prolonged odor exposure is pervasive across taxa and can be accompanied by the activation of xenobiotic responses.

## Introduction

Mosquitoes utilize a diverse array of molecular sensors to detect and respond to their external environment [1,2]. Of particular epidemiological importance are the olfactory receptors (ORs) which play a central role in both host discrimination and repellent avoidance [3,4]. The characterization of these molecular sensors may be of great value in the development and implementation of novel disease control and mitigation strategies. Despite this, with the exception of *Anopheles gambiae*, the olfactory receptor repertoire of most vector mosquitoes remain predominantly uncharacterized [5,6]. This in large part is due to the highly divergent nature of olfactory receptors which makes sequence-based characterization largely ineffective [7]. There are a few well-characterized receptors that are evolutionarily conserved across many species such as the Or8 clade [8]. This clade contains highly responsive and selective receptors for the "mushroom alcohol" odorant 1-octen-3-ol and its structural analogs [8–10].

relevant data are within the paper and its
Supporting Information files.

**Funding:** F.M., A.J.B., and M.D. were supported by
the National Institute of Allergy and Infectious
Disease (NIAID) of National Institutes of Health
(NIH) (1R21AI142140). B.E. was supported by The
Centers for Disease Control and Prevention (CDC),
Southeastern Center of Excellence in Vector-borne
Disease (U01CK000510). The funders had no role
in study design, data collection and analysis,
decision to publish, or preparation of the
manuscript. There was no additional external
funding received for this study.

**Competing interests:** The authors have declared
that no competing interests exist.

Transcriptional expression of the ORs is largely restricted to the antenna, maxillary palp and proboscis in insects [1,2,11–13]. Recently, a phenomenon was discovered in *Drosophila melanogaster* as well as murids like *Mus musculus* that prolonged exposure to an odorants is positively correlated with modulation of the mRNA transcript abundance of its cognate receptor in olfactory receptor neurons (ORNs) [14]. Further evaluation of this phenomenon in *Drosophila* concluded that modulation of mRNA could correctly predict receptor-ligand interaction 69% of the time using an aptly named strategy; Deorphanization of Receptors based on Expression Alteration of mRNA levels (DREAM) [15]. Considering the current landscape of deorphanization in insects, this strong predictive relationship presents an enticing strategy for conducting an initial high-throughput global screen to identify potential ligand-receptor interactions using RNA sequencing.

The mechanism driving this mRNA modulation caused by prolonged odor exposure is not well understood; it has been suggested to be a form of olfactory sensory adaptation and likely a ubiquitous feature of ORNs [16,17]. Since neurons need to respond to changes in chemosensory cues by reducing noise while remaining dynamically responsive to new and salient cues, modulation at the transcription level could potentially be an evolutionary response to sustained noise [18]. Alongside ORs, other gene families have been associated with the detection of chemicals in the environment include: Ionotropic Receptor (IRs), Gustatory Receptors (GRs), Odorant Binding Proteins (OBPs), Pickpocket channels (PPKs), and Transient Receptor Potential (TRPs) ion channels [1,2]. These gene families have the potential to be transcriptionally regulated by odor exposure.

Odors are volatile chemicals that could trigger xenobiotic response pathways. Insects have evolved xenobiotic response pathways capable of metabolizing, excreting, and reducing the penetration of exogenous chemical compounds [19]. These pathways include Insect Cuticle Protein (ICPs), Cytochrome P450 (CYPs), Glutathione S-transferase (GSTs), Glucuronosyl Transferases (GTs), Scavenger Receptor Type B (SRBs) and Carboxyl/cholinesterases (CCEs) [20–23]. Induced transcriptional responses of these gene families have been observed but has largely focused on responses to high concentration exposure to pesticides and toxins rather than what are considered behaviorally salient odors [24–26].

Here we observe that odor-evoked transcriptional changes are a recurrent property of olfactory systems by inducing it in *Aedes aegypti*. First, we show that 1-octen-3-ol odor exposure caused concentration-dependent, time-dependent, and temporary modulation with its best-known receptor, Or8. We then captured odor-evoked transcriptomes (OET) to globally investigate the response of prolonged exposure to 1-octen-3-ol in different gene families. The gene families known to directly interact with 1-octen-3-ol, ORs and OBPs, encompassed most chemosensory changes, with 24 ORs differentially expressed. The next most transcriptionally modulated genes were those involved in xenobiotic response with changes observed primarily in the CYP, ICP, and GT gene families. Overall, this study suggests that OETs may provide insight into odor-ligand receptor relationships, the role of xenobiotic responses in olfaction, and how odor-evoked transcriptional changes may allow for sensory adaptation.

## Results and discussion

### Or8 mRNA modulation is 1-octen-3-ol exposure dependent

Despite the differences in the structural organization of the olfactory system between *Mus musculus* and *Drosophila melanogaster*, prolonged exposure of odors resulting in alterations in transcripts abundance have been observed in both species [14,27,28]. This makes it reasonable to infer that odor-evoked transcriptional changes will be observed in other organisms given similar stimulus. We therefore hypothesized that *Aedes aegypti* OR transcripts will undergo

similar modulation when exposed to an odor. To test this hypothesis, we needed a candidate receptor whose response profile had been well-characterized for its strength and specificity of odor response. We selected *Ae. aegypti* Or8, a member of the Or8 clade of highly responsive and narrowly tuned receptors which primarily respond to 1-octen-3-ol and its analogs, as a model for investigation of these sensory responses [8–10].

Prolonged exposure to chemicals has the potential to cause acute toxicity in insects at varying proximity and concentration [29]. We therefore performed a no observable adverse effect level (NOAEL) in our custom odor exposure bioassay at varying concentrations of 1-octen-3-ol to determine the maximum concentration that mosquitoes can be exposed to in our assay without observable behavioral impairments at 6 and 24 hours (S1 Table). This was done so that any mRNA modulation is likely not caused by potential odor toxicity but rather as a response to odor detection. At 1% v/v concentration no observable impairment in females were observed at 6 and 24 hours compared to only mineral oil exposed mosquitoes.

To determine the effect odor exposure time and concentration on *Aedes Or8* transcript abundance, we designed a TaqMan qRT-PCR based assay (Fig 1A and S2 Table). To investigate the effect of exposure time, we performed a time course assay exposing female mosquitoes to 1% 1-octen-3-ol in mineral oil (treatment) or mineral oil alone (control) for 1, 3, 6, 12, and 24 hours. After 3 hours of 1-octen-3-ol exposure, the *Or8* levels began to significantly downregulate and remained suppressed at 6,12, and 24 hours of exposure time relative to their respective controls (Fig 1B). Next, we wanted to determine the effect of odor concentration on *Or8* mRNA relative abundance. Starting at 1% 1-octen-3-ol, we proceeded to incrementally decrease the concentration of the odor to $10^{-8}$ (Fig 1C). We found that *Or8* transcript levels were significantly reduced as low as a $10^{-7}$ dilution relative to their controls and the decrease was found to strongly fit a linear trend model (Fig 1C). In previous studies utilizing the DREAM approach the receptor response was found to be reversible when the odor was removed and the organism given sufficient time to recover [14]. To determine if the odor-dependent suppression of the *Or8* transcript was reversible, we exposed female mosquitoes to 1-octen-3-ol for 6 hours or 6 hours followed by 6 hours of recovery without 1-octen-3-ol odor. We observed a significant increase in Or8 mRNA when the odor was removed, and mosquitoes were allowed to recover for 6 hours (Fig 1D). Taken together, these assays show that modulation of *Or8* occurs in a temporal and concentration-dependent manner in *Ae. aegypti* when exposed to 1-octen-3-ol and indicates a similar transcriptional response profile for a given odor exposure as previously reported in *Mus musculus* and *Drosophila melanogaster* [14,15].

## An *Ae. aegypti* odor-evoked transcriptome

Olfactory genes are expressed in relatively low concentrations and largely restricted to the antenna, proboscis and maxillary palps in mosquitoes [1,2,12]. A large-scale exposure assay designed to detect these low expression transcripts resulted in a similar reduction in *Or8* as previously observed by qRT-PCR (Fig 2A). To ensure detection of low expression transcripts, the large-scale exposure assay uses approximately 120 female mosquitoes exposed to 1-octen-3-ol for 6 hours followed by removal of their sensory appendages (Fig 2B). To gain a global perspective of gene expression dynamics resulting from prolonged exposure, we performed Illumina RNA sequencing on female mosquitoes exposed to 1-octen-3-ol for 6 hours alongside unexposed controls. Together the 1-octen-3-ol exposed (OctT) and unexposed controls (OctC) resulted in more than 1.5 billion raw reads generated by Illumina sequencing with each treatment group comprised of three biological replicates. After quality control and trimming the total number of reads in the samples ranged from 206,372,686 to 291,393,839 (S3 Table). Those reads were then mapped to the *Aedes aegypti* AaegL5.3 annotated transcriptome using

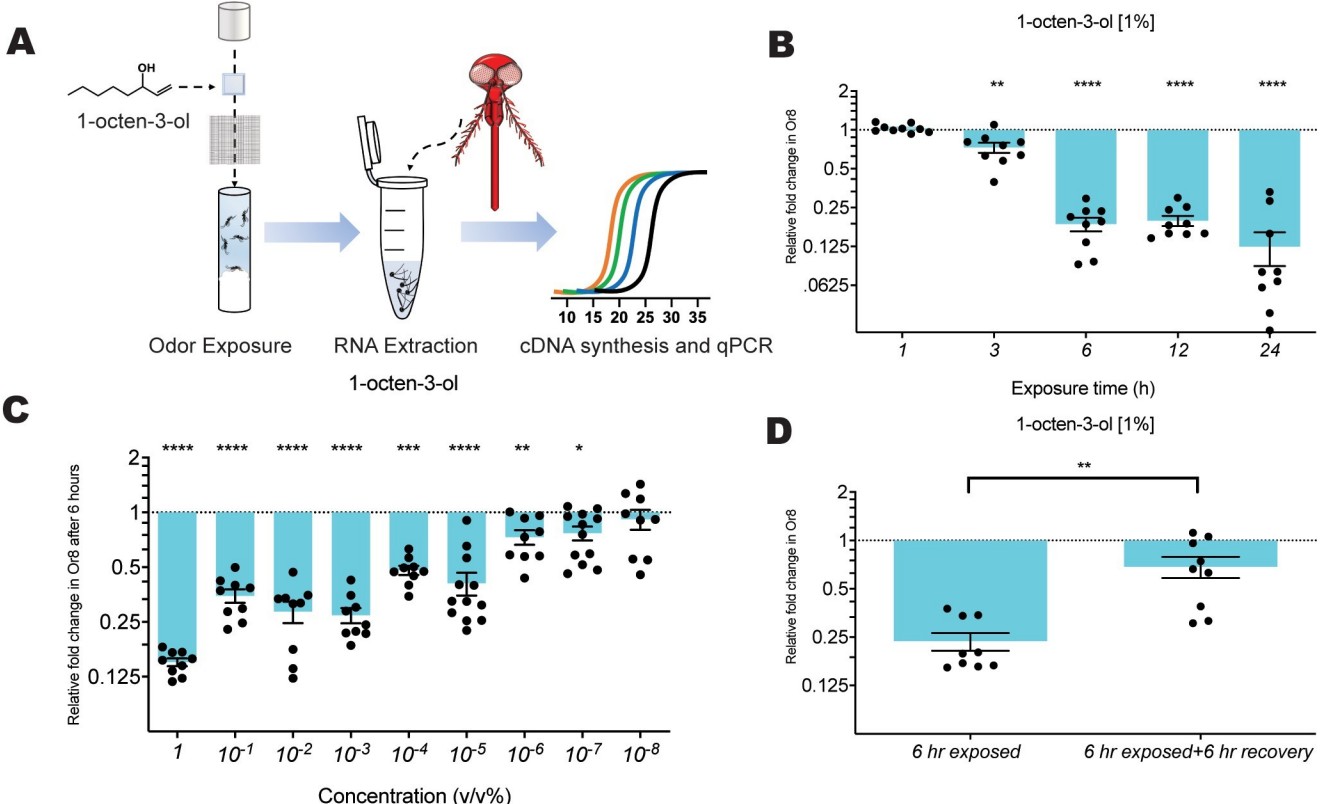

**Fig 1. Odor-evoked modulation of Or8.** (A) Diagram of small-scale exposure assay (B,C,D) Relative fold change in mRNA expression of OR8 evaluated using RT-qPCR from head tissue (B) when exposed to 1% 1-octen-3-ol for 1, 3, 6, 12, or 24 hours (C) when exposed to varying concentration of 1-octen-3-ol for 6 hours (D) when exposed to 1% 1-octen-3-ol in mineral oil for 6 hours and evaluated; then given an additional 6 hour unexposed before evaluation against their respective controls. Control mosquitoes for each condition were exposed to mineral oil alone at the associated time interval. Each biological replicate consists of five female mosquitoes, n = 3–4. mRNA level plotted on a 2-log scale as the mean ratio ± s.e.m. of mRNA levels compared to the mean levels of their respective controls and evaluated using two-tailed Mann-Whitney U test $^*P \leq 0.05$, $^{**}P \leq 0.01$ $^{***}P \leq 0.001$, $^{****}P \leq 0.0001$. One-way ANOVA linear trend analysis was done on the concentration assay, which had a $P \leq 0.0001$, with an alerting R-square = 86.07.

Salmon with an average mapping rate amongst the samples of 75.22%. Following mapping the DESeq2 differential gene analysis was conducted comparing OctT vs OctC. Observed fold-change ranged from 12-fold increase to 11-fold decrease, the smallest p-value observed was $10^{-43}$ (Fig 2C and S4 Table). The differential genes analysis showed 4.5% of genes were differentially expressed with 589 significantly downregulated and 245 upregulated (Fig 2D).

## Gene Ontology analysis

The differentially expressed genes (DEGs) identified by DESeq2 were analyzed using the Vectorbase Release 55 Gene Ontology Enrichment and Metabolic Pathway Enrichment tool. Of the 834 identified DEGs, 582 had at least one associated GO Term. Total Gene Ontology terms associated with the set of DEGs were categorized by biological process (530), cellular component (599), and molecular function (1131) (S5 Table). After reducing redundant terms, the analysis identified 126 biological process, 68 cellular component, and 33 molecular function GO Terms descriptions that were significantly represented amongst the DEGs. The DEGs with GO terms for biological processes were found to be involved in processes associated with sensory perception of smell (GO:0007608) and response to chemicals (GO:0042221). The cellular components involved were associated primarily with integral component of membrane

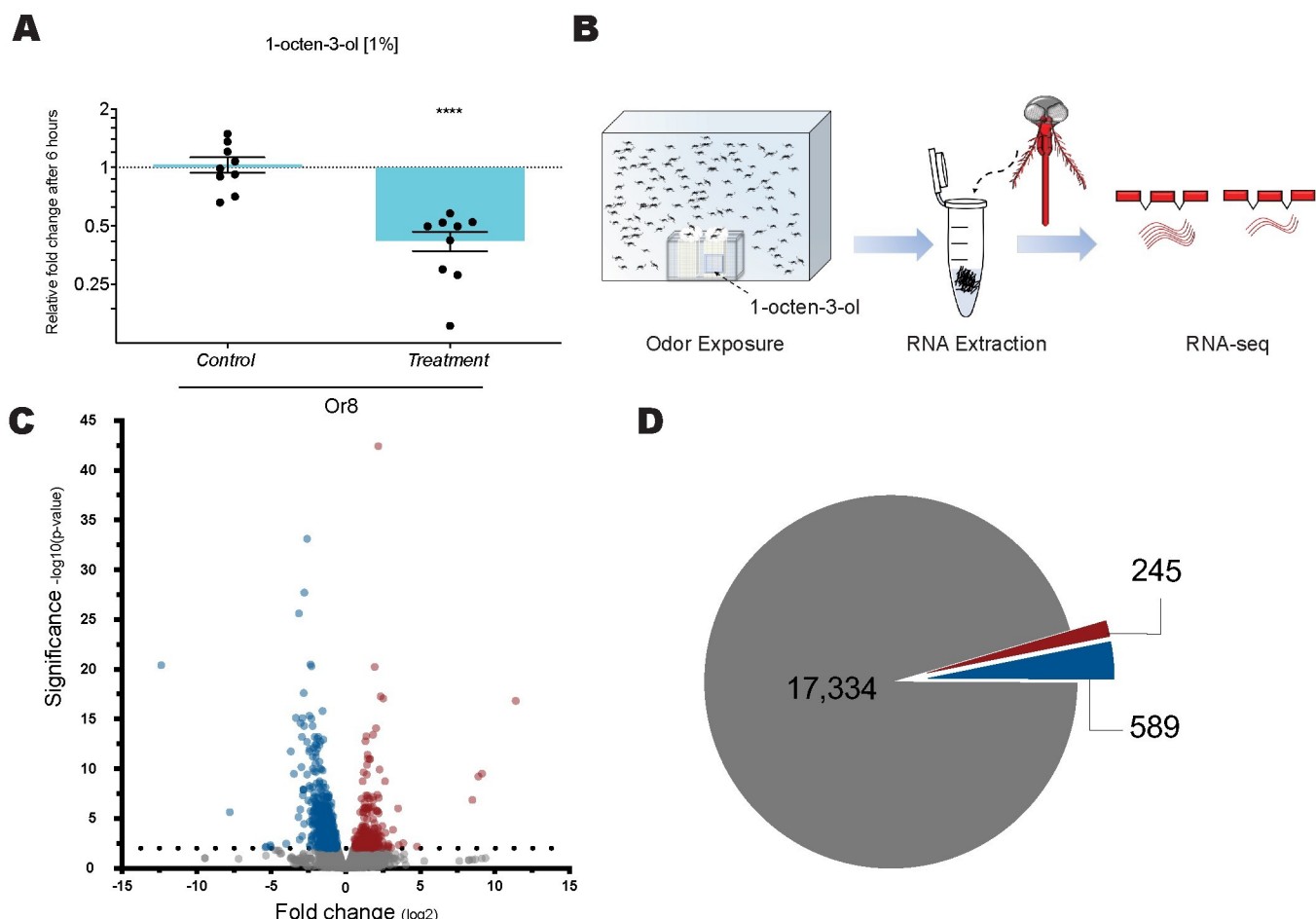

**Fig 2. Odor-evoked transcriptome of 1-octen-3-ol.** (A) qPCR validation of large-scale exposure assay; two-tailed Mann-Whitney U test, **** P ≤ 0.0001. (B) Mosquitoes are exposed to 1-octen-3-ol for 6 hours followed by dissection of chemosensory tissue for generation of RNA-seq libraries. (C) Volcano plot of all detected genes; downregulated genes represented by blue dots, genes upregulated represented by red dots, and genes with no transcriptional changes represented in gray, at P < .01. (D) Pie chart indicating downregulation of 589 genes (blue), upregulation of 245 genes (red), and no change in 17,334 genes (gray).

(GO:0016021) and finally the molecular function of the DEGs associated with GO Terms involved mainly oxidoreductase activity (GO:0016491), odorant binding (GO:0005549), and iron ion binding (GO:0005506) (Fig 3A).

To investigate the pathways involved in the response to 1-octen-3-ol exposure, we preformed KEGG Pathway analysis on the DEGs. These genes were associated with 3966 KEGG description comprising of 145 unique pathway terms (S5 Table). KEGG Analysis identified pathways significantly represented to include drug metabolism–cytochrome P450 (ec00982), retinol metabolism (ec00830), as well as arscorbate and aldarate metabolism (ec00053) (Fig 3B). Taken together, the functions and pathways identified in this analysis largely classified into two general groups: those involved in chemoreception and those primarily involved in xenobiotic-like responses. We see that the transcriptional effect of exposure is rather restrictive in scope and therefore likely either the results of direct depletion of transcripts or a feedback mechanism involving chemosensory and xenobiotic gene families.

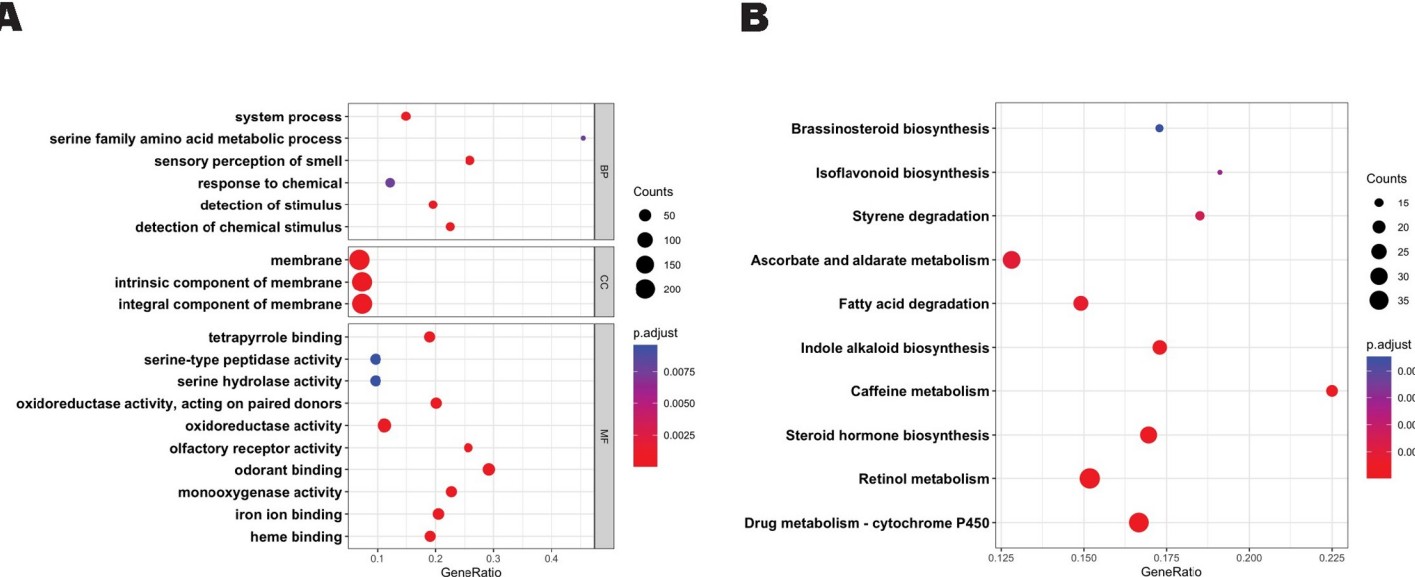

**Fig 3. Gene Ontology (GO) and KEGG pathways for DEGs.** (A) GO terms analysis for biological processes, cellular components, and molecular functions. (B) KEGG pathway analysis. The circle size represents number of genes, circle color represents padj. Gene ratio is the ratio of the enriched genes to the total number of genes in the relative pathway. P-values were adjusted by the Benjamini-Hochberg method.

## Differentially expressed genes associated with chemoreception

A comprehensive evaluation in *D. melanogaster* found modulation of mRNA could correctly predict either excitatory or inhibitory interaction 69% of the time [15]. We therefore wanted to know whether mRNA modulation could be utilized as initial screening tool to identify candidates potentially having an unknown ligand-receptor relationships in an *Ae. aegypti* background. In the initial DREAM study, modulation of ORs occurred only in terms of downregulation of receptors with none of the interacting receptors being upregulated in *Mus musculus* or *Drosophila melanogaster* [14]. Further evaluation in *Drosophila* found modulation as an indicator of ligand-receptor interaction corresponded with either an upregulation or downregulation event [15]. It is unclear whether this difference observed is simply properties of the odors used, the organism exposed, or differences in experimental set-up between the two studies.

In our global OET survey of the 97 olfactory receptors expressed at detectable abundance, 24 were downregulated at p < .01 with an average log2 fold-change of -1.75 (Fig 4A and S6 Table). Of the 24 ORs, Or8 was 11th, Or4 was 19th, and Or10 was 20th most downregulated. Or8, Or4 and Or10 are some of the best-characterized receptors in *Ae. Aegypti* and only Or8 responds strongly to 1-octen-3-ol while Or4 and Or10 respond to sulcatone, and indole, respectively [4,8,30]. Here, Or4 and Or10 represent false positives which suggests that OET data is not always a predictive indicator of odor-ligand receptor relationships. The remaining receptors are currently orphaned with unknown response repertoires, however, earlier work has shown that Or88, Or107, Or114, and Or115 are differentially expressed between male and female mosquitoes and have therefore been proposed to play important roles in regulating host-seeking behavior [31]. Expression of the 24 ORs identified have been found to be largely restricted to the antenna, apart from Or8 which is primarily found in the maxillary palp, and Or4 whose expression was comparatively high in rostrum tissue as well as antennal tissue [12]. The overall trend of all ORs was towards a downregulation event of transcript abundance. Of note is that Orco, the obligate co-receptor of ORs was downregulation by a log2 fold-change of

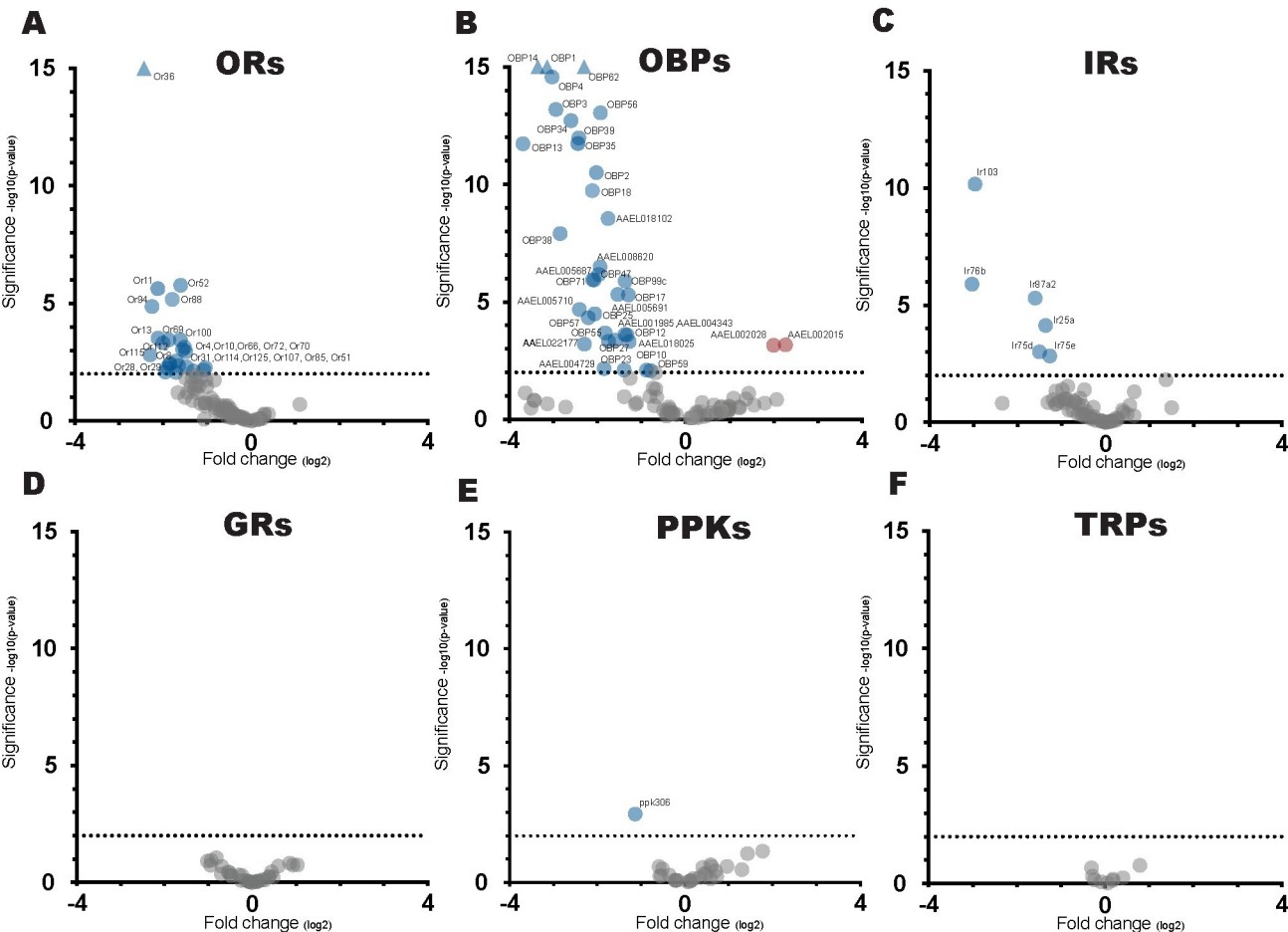

**Fig 4. Volcano plot of transcriptional changes of sensory perception gene families.** (A-F) Volcano plots of Olfactory Receptors (ORs), Ionotropic Receptors (IRs), Gustatory Receptors (GRs), Odorant binding proteins (OBP), Pickpocket (PPKs), and Transient receptor potential channels (TRPs). Genes downregulated at P < .01 represented in blue, genes upregulated at P < .01 represented in red, and genes with no transcriptional changes at P >.01 represented in gray, gene outside of scale represented as triangles.

-1.26 and that the 73 ORs above p >.01 averaged a log2 fold-change of– 0.52. Of the ORs not differentially expressed, evaluation in *Drosophila* suggests that some ligand responders will not differentially express after odor exposure, this would constitute false negatives and will go unidentified when using a DREAM or OET screening approach [15].

Efforts to understand the expression patterns of ORs indicate a much more complicated organization pattern than the canonical one-receptor one-neuron [32,33]. Examination of cell clusters from sensory appendages finds robust co-expression of ORs. If co-expressed ORs operate under similar regulatory mechanisms is still unclear, however amongst the differentially expressed ORs in our OET study, Or88-Or36, Or36-Or52-Or72, Or72-Or115-Or114, and Or100-Or94 were found to be co-expressed [33]. Understanding whether co-expression results in odor-gated ion channels composed of multiple tuning ORs with different ligand-specific regions could be helpful in the interpretation of our results [34].

Amongst all Diptera, 15 receptors have been identified that exhibit strong activation response to 1-octen-3-ol [6,8,10,35–39]. To further determine the viability of the 24 potential candidates identified through RNA-seq analysis we endeavored to compare them against known 1-octen-3-ol responding Dipteran receptors. Amongst Dipterans, sequence alignment

comparisons of olfactory receptors largely falls into the "twilight zone" in which sequence similarity is such that it can be difficult to distinguish between protein pairs of similar and non-similar structure [40–42]. Recent, cryo-electron microscopy studies of the *Machilis hrabei* Or5 found that the transmembrane region of the protein's s2-s6 helixes formed a simple binding pocket which recognized odor-ligands using weak intermolecular interactions [43]. This would suggest that the ligand preference is largely determined by the pocket's geometric shape formed by relationships of multiple helices. Three-dimensional structural comparison should provide insight on possible similarity of pocket formations independent of specific interacting residues. We therefore generated Alphafold2 predicted protein structure to allow for 3D structural comparison of our 24 candidates against the 15 known 1-octen-3-ol dipteran receptors using LGA for protein comparisons [44] (Fig 5A and S7 Table). Amongst the current orphaned receptors, the highest structural similarity score of 96.126 was between AaOr88 and AalOr88, which are predicted to share a similarly shaped binding pocket (Fig 5B and 5C). Members of the Or8 clade of specialized 1-octen-3-ol mosquito receptor shared high structural similarities to AaOr28, with a maximum similarity score of 90.069 with TaOr8. Another candidate of interest was AaOr13 which shared high structural similarity to AgOr4, AgOr20, and CqOr1 with scores of 86.43, 87.26, and 89.13, respectively. High structural similarity to known 1-octen-3-ol receptors provide a secondary indicator of support of potential ligand-receptor pairing.

Alongside excitation, inhibition has been implicated in the induction of mRNA modulation in odor exposure assay [15]. Inhibitory responses in studies, however, provide much lower response resolution when compared to excitation and therefore should be approached with more caution when being used to draw inferences [13,37]. In *An. gambiae* approximately 15 receptors show levels of inhibitory response in the empty neuron system [6]. Taken at face value, if similar excitation vs inhibition ratios exist in *Ae. aegypti*, then we would expect that many putative candidates could exhibit inhibitory responses when challenged with 1-octen-3-ol. Odor perception has been proposed to be combinatorial in nature with neuron activation or inhibition capable of encoding information, it therefore may be of interest to study the logic underlying inhibitory responses.

Outside of ORs of the chemosensory gene families, only OBPs have been shown to directly interact with alkenyl alcohols like 1-octen-3-ol [9,45]. Of the 137 genes with OBP annotations, 37 were significantly downregulated, with 2 OBP/ejaculatory bulb-specific genes upregulating at p < .01 (Fig 4B). OBPs function in olfaction appear to be versatile; they are implicated in diverse roles such as the transport of odor molecules, sensitivity modulation, and facilitating of odor degradation [46]. Modulation of ORs has been suggested to result in changes in sensory perception; the depletion of OBPs could potentially change olfactory perception by reducing the amount of odor transported through the lymph to olfactory receptors [47].

Within the Ionotropic Receptors (IRs), a small number of genes were significantly downregulated, most notably the Ir25a, and Ir76b co-receptors (Fig 4C). While Ir25a and Ir75b have functional roles in acid and amine reception, it is unknown whether any of the ligand-specific tuning receptors associated with them respond to 1-octen-3-ol [48,49]. Recent analyses shows that many neurons co-express multiple chemosensory receptor genes in *Ae. aegypti* and *Drosophila* including Ir76b in Or8-expressing B cell of the maxillary palp. [32,33]. How activation of a particular receptor effects the transcriptional regulation of other co-expressing receptors when stimulated is currently unknown.

Amongst the remaining sensory perception gene families GRs, TRPs, PPKs remained virtually unchanged with only one DEG amongst the three families (Fig 4D–4F). Unlike ORs, IRs, and OBPs, they are not believed to act as receptors for 1-octen-3-ol. This suggests that the mechanism driving mRNA modulation of ORs requires odor-ligand mediated activation of the sensory neuron.

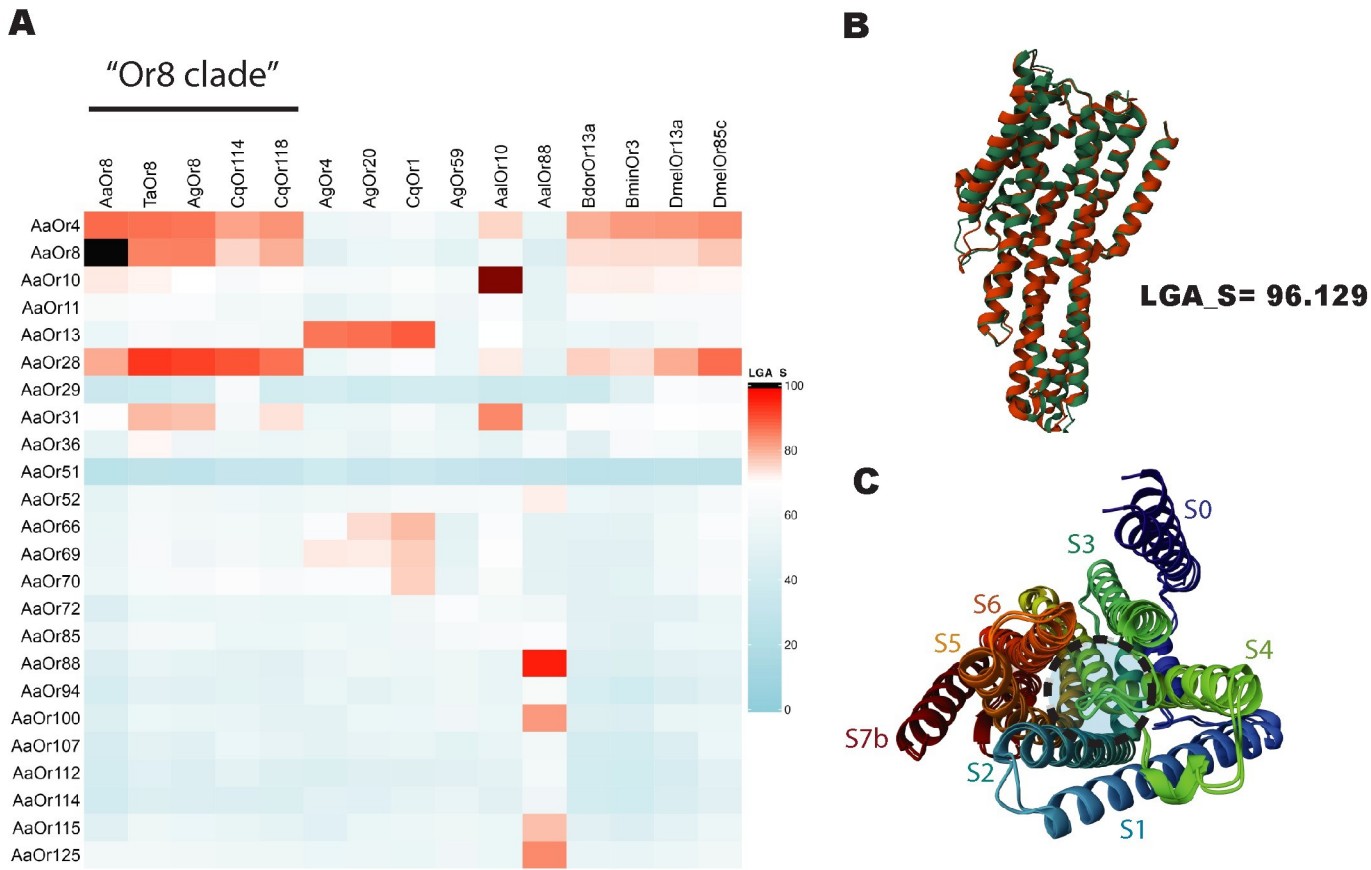

**Fig 5. Structural comparative analysis of candidate receptors with known 1-octen-3-ol receptors.** A. Heatmap of structure similarity score (LGA_S) values of candidate receptors superimposed onto known dipteran 1-octen-3-ol receptors generated using LGA program. Species shorthand: *Ae. aegypti* (AaOr), *Ae. albopictus*, (AalOr), *An. gambiae* (AgOr), *C. quinquesfasciatus* (CqOr), *T. amboinensis* (TaOr), *B. dorsalis* (BdorOr), *B. minax* (BminOr), *D. melanogaster* (DmelOr). B. Visualization of Alphafold2 generated 3D protein structures models of AaOr88 (orange) superimposed onto AalOr88 (green) along with corresponding structure similarly score. C. Rainbow palette of AaOr88 showing predicted binding pocket formed by its transmembrane helices, general area of pocket formed indicated by black dash-lined and transmembrane domains labeled S0 (purple) - S7b (red).

## Xenobiotic response genes are differentially expressed after odor exposure

Xenobiotic associated gene families include cytochrome P450 (CYPs), insect cuticle protein (ICPs), glucuronosyltransferases (GTs), glutathione S-transferase (GSTs), scavenger receptor type B (SRBs) and carboxyl/cholinesterase (CCEs) [20–23]. Exposure to sub-lethal concentrations of xenobiotic compounds such as fluoranthene, permethrin, and copper sulfate has been shown to induce changes in expression in members of these families. While volatile 1-octen-3-ol levels used were not high enough to cause any impairment in standing, walking, and flying in our assay, prior work in other insects has demonstrated larval toxicity at sufficiently high concentrations whereas similar doses are not toxic to *Aedes* larvae [50]. While we took care to use a dosage which would not induce observable behavioral impairment, this exposure is apparently sufficiently high enough to induce a xenobiotic response. Of the 152 cytochrome P450 members, 5 were upregulated and 27 downregulated (Fig 6A and S8 Table). Cytochrome induction by xenobiotic chemicals has been associated with insecticide tolerance undergoing transcriptional modulation as a result of exposure [22,24]. Of the remaining xenobiotic responsive families, insect cuticle proteins and glucuronosyltransferases had the most differentially expressed members followed by glutathione S-transferases, carboxyl/cholinesterase,

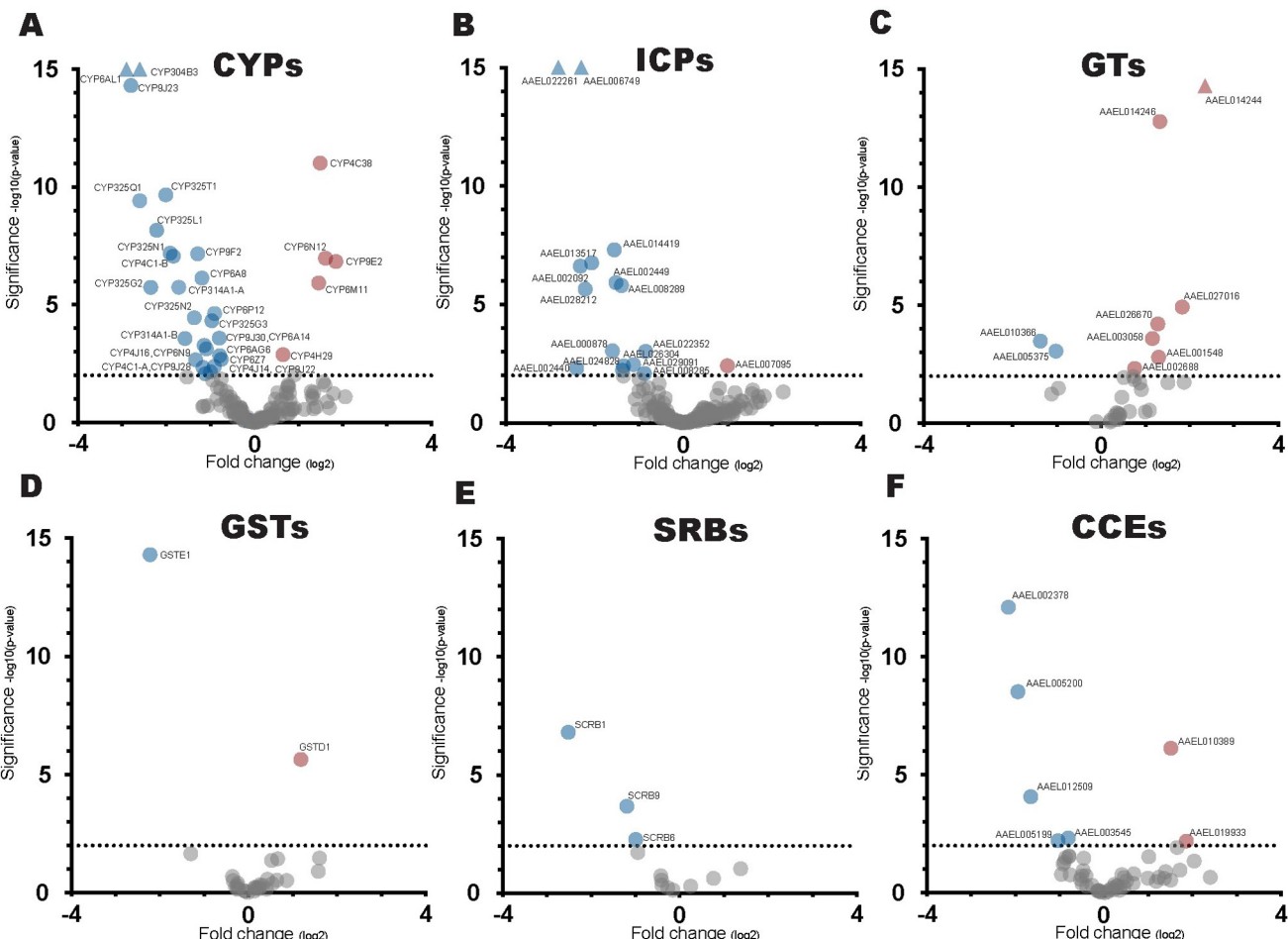

**Fig 6. Volcano plots of transcriptional changes of xenobiotic metabolism gene families.** (A-F) cytochrome P450 (CYPs), insect cuticle proteins (ICPs), glucuronosyl transferases (GTs), glutathione S-transferase (GSTs), scavenger receptor type B (SRBs), carboxyl/cholinesterase (CCEs), Genes downregulated at P < .01 represented in blue, genes upregulated at P < .01 represented in red, and genes with no transcriptional changes at P >.01 represented in gray, gene outside of scale represented as triangles.

scavenger receptor type B (Fig 6B–6F). Considering the broad transcriptional changes in xenobiotic response genes, it seems likely that xenobiotic modulation is a feature associated with prolonged odor exposure and could be used to assess whether odor application was effective when using this technique.

Most investigations into olfactory receptor responsiveness are "blind deorphanizations", that is, attempted without knowledge or prediction of potential receptor-ligand relationships. Generation and evaluation of an odor-evoked transcriptome induced by 1-octen3-ol exposure in *Ae. aegypti* demonstrated that prolonged exposure to one odorant could cause complex alterations in gene expression, some which may shed light on the regulatory architecture of the peripheral olfactory system. Our dataset however identified several known false positives if used as a sole indicator of ligand-receptor relationships, with the possibility of several undetermined false negatives. The limitations of this strategy as an initial screening tool to drive hypothesis-driven deorphanization warrant further investigation. Given its conservation across taxa, odor-evoked transcript responsiveness is likely an inherent property of olfaction and further study of this phenomenon may provide insight into the mechanisms underlying odor adaptation.

## Material and methods

### Statement of research ethics

All research was conducted in compliance with National Institutes of Health and Florida International University Environmental Health and Safety guidelines. Biohazard disposal, laboratory practices, facilities, and equipment were reviewed and approved by the Florida International University Institutional Biosafety Committee (IBC-21-022-AM03).

### Mosquito rearing

Orlando strain *Aedes aegypti* mosquitoes were maintained at 25–28 C with 70–80% relative humidity under a 14 hour light:10 hour dark cycle with light starting at 8 AM. Eggs were hatched in deoxygenated, deionized water containing pestle-crushed Tetramin tropical fish tablets (Catalog #16152, Tetra, Melle, Germany). Larvae were cultured in 2 L deionized water and fed Tetramin tablets in pans containing 200–250 larvae until they reach pupae stage. Pupae were allowed to emerge into BugDorm-1 Insect Rearing Cage (Catalog #1452, Bioquip, Rancho Dominguez, CA, USA) with a density of 200–250 mosquitoes per cage and given unlimited access to 10% sucrose solution. Mosquitoes were blood-fed using defibrinated sheep's blood (Catalog # R54016, Fisher Scientific, Waltham, MA, USA) using a heated glass feeder to generate eggs.

### Odorant exposure

Small scale odor exposure for downstream qRT-PCR analysis utilized *Drosophila* food vials (Catalog #32-117BF, Genesee Scientific, San Diego,CA, USA). Food vials with a cottonball (Catalog # 22-456-883, Thermo Fisher Scientific, Waltham, MA, USA) at the bottom were filled with 12 ml of deionized water. Cotton balls were flattened using a smaller sized food vial and any excess water decanted. Five female mosquitoes 7–12 days post-emergence were then aspirated into vials and mesh secured over the opening using a rubber band. 1-octen-3-ol (C. A.S. 3391-86-4) at the concentration of interest was mixed in mineral oil (v/v%) and vortexed. Sachet sleeves were made using Uline Poly tubing (Model No. S-3521, Uline, Pleasant Prairie, WI, USA) cut into 2 x 2 cm pieces and heat sealed at one end using a plastic film sealer. Sachets were then filled by pipetting 500 µl of mineral oil (Catalog # O122-1, Thermo Fisher Scientific, Waltham, MA, USA) or mineral oil and 1-octen-3-ol at described concentrations before sealing completely. Exposure was performed one group at a time to prevent contamination. The sachet was placed flat on top of mesh and the foam vial plug added with care to prevent rupture of the sachet from over compression. After elapsed time mosquitoes, the mosquitoes were examined for mortality and activity. Tubes were placed on ice to anesthetize the live mosquitoes. Whole heads were removed using a dissecting microscope and tweezers and placed in in 200 µl of RNAlater stabilization solution (Catalog #AM7020, Invitrogen, Carlsbad, CA, USA). Tubes were then centrifuged at 10,000 g for 30 secs and transferred to -80˚C freezer.

Large scale odor delivery occurred in a modified 10.25" x 8.5" x 2.375" insect habitat (Item # 6130, Kristal Educational, Quebec, Canada). Odor Sachets were made by pipetting 2000 µl of mineral oil alone or mineral oil and 1-octen-3-ol into 3.81 cm x 3.81cm made using Uline Poly tubing (Model No. S-3521, Uline, Pleasant Prairie, WI, USA) and heat sealing using a plastic film sealer. Sachets were placed in cube modular test tube rack and covered in mesh to prevent direct physical contact. A rack containing the odor sachet was placed in the center of the insect habitat and the lid then sealed. An aspirator was then used to add 120 female mosquitoes into the sealed container via an auxiliary port on the side of the habitat. After the elapsed time, mosquitoes were examined for mortality and activity. A dissecting microscope and tweezers were

then used to remove antennae, maxillary palps, and proboscises and immediately preserved. For qRT-PCR experiments, samples preservation was accomplished by snap-freezing using a dry ice-ethanol bath. All tissue samples used for downstream transcriptome profiling via RNA-seq were snap frozen in an ethanol dry ice bath. Tissue samples were stored in a -80˚C freezer till extracted.

No Observed Adverse Effect Level (NOAEL) assay was conducted using the same experimental set-up as previously described for small scale odor exposure. Mosquitoes were exposed at 4%, 2%, 1% and 0.5% as well as a no odor mineral control in triplicate. The ability of the mosquitoes to stand, walk, and fly were recorded at 6 and 24 hours and then classified as either alive, moribund, or dead according to WHO guidelines for chemical spatial testing [51].

## RNA extraction

RNA extractions were conducted in parallel using a guanidinium thiocyanate-phenol-chloroform extraction method. Mosquito tissues were suspended in a 1 mL solution containing 4 M guanidine thiocyanate (CAS: 593-84-0), 0.5% Sarkosyl (CAS: 137-16-6), chloroform (CAS: 67-66-3), and 0.1 M 2-mercapthoethanol (CAS: 60-24-2). Using RNase-free disposable pellet pestles (Catalog #12-141-364, Thermo Fisher Scientific, Waltham, MA, USA) the samples are then manually homogenized until no visible structures remain. After tissue homogenization, the samples were extracted twice with phenol-chloroform. RNA was purified from the recovered aqueous solution using the RNAid Kit supplied by MPBio (catalog #111007200). RNA-MATRIX beads were used to bind RNA using 5 µl for 5 whole heads and 10 µl for 120 sensory tissues. The beads were then washed twice using RNA Wash Concentrate to remove any remaining containments before eluting in 20 µl DEPC-treated water. Sample concentration and quality were determined using Thermo Scientific NanoDrop 2000c (Thermo Fisher Scientific, Waltham, MA, USA).

## cDNA synthesis and qRT-PCR

DNA contamination was then removed from RNA samples using Turbo DNA-free removal Kit (Catalog #AM1906, Invitrogen, Carlsbad, CA, USA) following manufacturer's instructions. Reverse transcription preformed using Verso cDNA Synthesis Kit (Catalog AB1453A, Thermo Fisher Scientific, Waltham, MA, USA) using oligo(dT) to generate cDNA libraries as per manufacturer protocol. Total RNA input for each cDNA synthesis reaction was normalized to a mass of 300 ng RNA in a total reaction volume of 20 µl. Following cDNA synthesis, samples were treated with RNase Cocktail Enzyme Mix (Catalog #AM2286, Thermo Fisher Scientific, Waltham, MA, USA). To assess cDNA quality and lack of gDNA amplification, samples were amplified using Amplitaq360 master mix (Catalog #4398901, Thermo Fisher Scientific, Waltham, MA, USA) with ribosomal protein L32 primers (S2 Table) in an endpoint reaction (10 min at 95˚C; 30 s at 95˚C, 1 min at 72˚C, 35 cycles; 72˚C for 5 min) and were run on an agarose gel. Only the expected cDNA product was observed for all samples; no longer amplificons originating from gDNA were observed, indicating lack of gDNA contamination as the primer pair spans an intron. Each gene of interest was amplified using a TaqMan Gene Expression Assay reaction consisting of two sequence-specific PCR primers with a custom TaqMan probe (Catalog #4331348, Thermo Fisher Scientific, Waltham, MA, USA) in 2X Universal Master Mix (Catalog #4324018, Thermo Fisher Scientific, Waltham, MA, USA). The TaqMan assay was performed in technical replicate for each biological replicate. RT-qPCR was performed on a 7500 Real Time PCR System from Applied Biosystems using universal cycling conditions (10 min at 95˚C; 15 s at 95˚C, 1 min 60˚C, 40 cycles). Ribosomal protein L32 was used as endogenous control to normalize variation in total cDNA between samples. Raw data output analyzed

using SDS v1.5.1 software with detection threshold set at = 0.2. Technical replicates with Ct values higher than 0.5 Ct from their nearest technical replicate were classified as outliers and discarded. Relative fold-change calculated by $2^{-\triangle\triangle Ct}$ method using an in-house Excel macro as previously described [52].Statistical analysis of fold-change data conducted using GraphPad Prism 8 software package (GraphPad Software, San Diego, CA, USA).

## Library preparation and RNA-sequencing

RNA library preparation and sequencing were conducted at GENEWIZ, LLC (South Plainfield, NJ, USA). RNA sample integrity and quantification was assessed using TapeStation (Agilent Technologies, Palo Alto, CA, USA) and Qubit 2.0 Fluorometer respectfully (Invitrogen, Carlsbad, CA, USA). NEBNext Ultra RNA Library Prep Kit was then used to prepare sequencing libraries according to manufacturer's instructions (NEB, Ipswich, MA, USA). Oligo(dT) beads were used to enrich mRNA from 200 ng total RNA prior to fragmenting for 15 minutes at 94˚C. After fragmentation cDNA first and second strand were synthesized. cDNA fragments were end repaired and adenylated at 3'ends, and universal adapters were ligated to cDNA fragments, followed by index addition and library enrichment by PCR with limited cycles. Agilent TapeStation (Agilent Technologies, Palo Alto, CA, USA), Qubit 2.0 Fluorometer (Invitrogen, Carlsbad, CA, USA) as well as quantitative PCR (KAPA Biosystems, Wilmington, MA, USA) were used to validate and quantify the sequencing libraries.

Libraries were sequenced across three Illumina HiSeq lanes with 2x150 bp Paired End chemistry according to manufacturer's instructions. HiSeq Control Software (HCS) were used to conduct base calling and Image analysis and Illumina HiSeq Raw sequence data (.bcl files) were converted into fastq files and de-multiplexed using Illumina software. Index sequence identification allowed for one mismatch.

## Quality control and read mapping

Quality control of raw reads evaluated by base sequence quality analysis and base composition analysis using FASTQC v0.11.5. Raw reads Adapters were removed using Trimmomatic v0.36. AaegL5.3 annotated transcript containing 29,025 mRNA, 4,155 LncRNA, 384 and was downloaded from Vectorbase (release 55). Trimmed read pairs aligned to annotated transcript set from reference genome AaegL5.3 using Salmon v9.1.

## Differential transcription analysis

Raw transcript read counts output from Salmon were summarized to gene read counts using tximport v1.18.0 R/Bioconductor package [53]. Differential gene expression analysis based on negative binomial distribution was then conducted using DESeq2 v1.34.0 package in R Studio v1.2.1335 comparing 1-octen-3-ol exposed and unexposed gene counts [54]. Genes were considered significantly different if they had at an adjusted p-value $\leq$ 0.01. To identify genes in the chemosensory and xenobiotic families, Vectorbase accession number were used to retrieve Vectorbase descriptions, PFAM descriptions, and Interpro descriptions using Vectorbase Release 55 search function. Gene families gensets were then manually curated based on available descriptions. Volcano plots of total genes and selected gene families were then generated using GraphPad Prism 8 software package using log2 Fold change and adjusted p-values.

## GO enrichment analysis and KEGG pathway analysis

The identified differentially expressed genes (DEGs) were uploaded to Vectorbase Release 55 for KEGG pathway analysis and Gene Ontology analysis in *Aedes aegypti* LVP_AGWG. The

Vectorbase GO enrichment tool was used to determine biological process, cellular component, and molecular function enrichment using both computed and curated evidence any redundant terms were reduced using Revigo (http://revigo.irb.hr/) [55]. The Vectorbase Metabolic Pathway Enrichment tool was then used for KEGG enrichment. Significance was determined using Benjamini-Hochberg corrected p values of $\leq 0.01$.

## Protein structure comparison using Alphafold2 generated models

Protein sequences were obtained from Vectorbase with the exception of TaOr8 which was taken from the initial deorphanization study [8]. Protein modeling was conducted using ColabFold:Alphafold2 using MMseq2 online server [56]. The best ranked model was then selected for downstream comparison. LGA (Local-Global Alignment) method software was used for alignments of Alphafold2 generated structures. Selected option used were "-4 -sia" in which the best superposition (according to the LGA technique) is calculated completely ignoring sequence relationship ("-sia") between the two proteins, and the suitable amino acid correspondence (structural alignment) is reported. Structure similarity score (LGA_S) is then extracted from each output and put into a data matrix. Heatmap of similarity scores created using ggplot2 3.4.0 and Complex Heatmap 2.13.1 packages in R Studio v1.2.1335. Superimposed protein structure image composed using Mol 3D Viewer online tool available through RSCB Protein Data Bank.

## Supporting information

**S1 Table. No observed adverse effect level assay at 6 hours and 24 hours of 1-octen-3-ol exposure.**
(XLSX)

**S2 Table. Relative fold change values of Or8 mRNA levels evaluated using qRT-PCR probes in time course, concentration curve, recovery, and validation assays.**
(XLSX)

**S3 Table. RNA-seq mapping statistics and gene-level read counts.**
(XLSX)

**S4 Table. DESeq2 differential gene expression analysis 1-octen-3-ol exposed vs control.**
(XLSX)

**S5 Table. Gene Ontology (GO) and KEGG pathway associated terms and summary.**
(XLSX)

**S6 Table. Sensory perception gene families (ORs, IRs, GRs, OBPs, PPKs, TRPs) DE analysis evaluated using DESeq2.**
(XLSX)

**S7 Table. Structure similarity scores (LGA_S) of Alphafold2 generated candidate receptors superimposed onto known dipteran 1-octen-3-ol receptors.**
(XLSX)

**S8 Table. Xenobiotic metabolism gene families (CYPs, ICPs, GTs, GSTs, CCEs, SRBs) DE analysis evaluated using DESeq2.**
(XLSX)

**S9 Table. Key resources.**
(XLSX)

## Acknowledgments

We would like to thank Dr. Marcela Nouzova and Dr. Andre Luis Da Costa Da Silva for offering helpful suggestions and insight during this process. We would also like to thank the Fernando Noriega's laboratory for sharing their knowledge and equipment with us.

## Author Contributions

**Conceptualization:** Fredis Mappin, Matthew DeGennaro.

**Data curation:** Anthony J. Bellantuono.

**Formal analysis:** Anthony J. Bellantuono.

**Funding acquisition:** Matthew DeGennaro.

**Investigation:** Fredis Mappin, Babak Ebrahimi.

**Supervision:** Matthew DeGennaro.

**Visualization:** Matthew DeGennaro.

**Writing – original draft:** Fredis Mappin.

**Writing – review & editing:** Fredis Mappin, Anthony J. Bellantuono, Matthew DeGennaro.

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
