## [Decision Letter · Decision Letter 0]

17 Jul 2023

PONE-D-23-09227Odor-evoked transcriptomics of Aedes aegypti mosquitoesPLOS ONE

Dear Dr. DeGennaro,

Thank you for submitting your manuscript to PLOS ONE. After careful consideration, we feel that it has merit but does not fully meet PLOS ONE’s publication criteria as it currently stands. Therefore, we invite you to submit a revised version of the manuscript that addresses the points raised during the review process.

Both reviews and myself found the data convincing and the manuscript well-presented. However, I must agree with reviewer 2 that limitations of the approach must be thoroughly discussed and towards that end I recommend you follow  and throughly reply too these suggestions.

We look forward to receiving your revised manuscript.

Kind regards,

Efthimios M. C. Skoulakis, PhD

Academic Editor

PLOS ONE

Journal Requirements:

"F.M., A.J.B., and M.D. were supported by the National Institute of Allergy and Infectious Disease (NIAID) of National Institutes of Health (NIH) (1R21AI142140-01).The funders had no role in study design, data collection and analysis, decision to publish, or preparation of the manuscript."

Please respond by return e-mail so that we can amend your financial disclosure and competing interests on your behalf.

5. We note that Figures 1 and 2 in your submission contain copyrighted images. All PLOS content is published under the Creative Commons Attribution License (CC BY 4.0), which means that the manuscript, images, and Supporting Information files will be freely available online, and any third party is permitted to access, download, copy, distribute, and use these materials in any way, even commercially, with proper attribution. For more information, see our copyright guidelines: http://journals.plos.org/plosone/s/licenses-and-copyright.

a. You may seek permission from the original copyright holder of Figures 1 and 2 to publish the content specifically under the CC BY 4.0 license. 

Reviewers' comments:

Reviewer's Responses to Questions

**Comments to the Author**

1. Is the manuscript technically sound, and do the data support the conclusions?

Reviewer #1: Yes

Reviewer #2: Partly

2. Has the statistical analysis been performed appropriately and rigorously? 

Reviewer #1: Yes

Reviewer #2: Yes

3. Have the authors made all data underlying the findings in their manuscript fully available?

Reviewer #1: Yes

Reviewer #2: Yes

4. Is the manuscript presented in an intelligible fashion and written in standard English?

Reviewer #1: Yes

Reviewer #2: Yes

5. Review Comments to the Author

Reviewer #1: The authors have assessed the feasibility of using single odorant exposure and subsequent differential expression analyses of control vs. exposed olfactory tissues to identify a cognate olfactory receptor(s) in an important vector of human disease. The project is well designed and data accurately reported. This work is easily worthy of publication in PLoS One and will draw interest from many researchers in the field of chemical ecology. My main criticism is that it is uncertain whether this approach is broadly useful for determining cognate receptors. I agree that the approach appears to narrow the search, and that when paired with protein structure modelling, the set of candidate genes becomes even smaller. On the other hand, there are still many potential genes in that candidate set and it is unclear whether these results would be similar for broadly tuned ORs. It may be more appropriate to steer the conclusion towards the broad array genes with altered expression as uncovering an unappreciated feature of neurosensory tissues.

Part 1 of the experiment exposes females Aedes aegypti to 1% 1-octen-3-ol for varying lengths of time and concentration to optimize effect, with 6 hours chosen as an ideal length of exposure. The researchers measured OR8 expression via qRT-PCR. This optimization was important before investing in full transcriptomes. The n-values reported were appropriate. The “no observable adverse effect level” NOAEL control was a good choice. A 1% odor concentration was well chosen for this study.

Part 2 exposes experimental females to the odorant and other females to control for 6 hours followed by RNA-seq in three biological replicates per condition. 4.5% of genes were differentially expressed. This type of broad genetic shift makes identifying cognate receptors more difficult. The functions and pathways identified in this analysis largely classified into two general groups: those involved in chemoreception and those primarily involved in xenobiotic-like responses. The rationale for focusing on these two sets was well reasoned.

Part 3 conducts a structural analysis of binding pocket similarity (Alphafold2 predicted protein structure) to determine if the candidate receptor list could be trimmed further to identify the true receptor-ligand pairings. These experiments seem exhaustive given current technological limitations. Heterologous expression of single receptors or sets of receptors or live calcium imaging would be laborious and time-consuming extensions of this work.

Specific comments:

Line 342 - check wording

Line 182 - "Differentially expressed genes associated with chemoreception"

In the first paragraph in this section, it may be better to discuss why OR8 was not the most depleted OR. Which other ORs are known to be co-expressed with OR8 in specific sensory neurons in Aedes? Perhaps the other depleted ORs you have identified are simply the result of differential neuronal activity over the 6 hours?

Line 284 - “Generation and evaluation of an odor-evoked transcriptome induced by 1-octen3-ol exposure in Ae. aegypti support its role as an initial screening tool for identifying potential ligand-receptor pairings.”

My suggestion would be to rephrase this final conclusion slightly. This technique may or may not be a useful initial screening tool based on the data presented here. Its usefulness is dependent on whether or not the 24 OR candidates identified here have some functional relationship with the odorant 1-octen3-ol. OR transcript depletion could be an indirect result of 1-octen3-ol-evoked activity in a subset of ORNs. Further, RNA-seq is somewhat expensive in non-model organisms. Even still, the expression pattern revealed here is fascinating and could point to biologically relevant changes occurring downstream of odorant exposure. I think the take-home message should be that long exposures to single odorants lead to complex changes in gene expression, some of which may reveal the genetic architecture of the peripheral olfactory system.

Even in its current form, I believe this to be worthy of publication. The paper is very well written and the data is presented transparently and without agenda. This is great work. I leave it to the editor to decide whether the main conclusion should be revisited.

Reviewer #2: The authors present work using the DREAM (Deorphanization of receptors based on expression alterations of mRNA levels) technique to identify potential receptors that respond to the ecologically important odorant 1-octen-3ol in Aedes mosquitoes. Using qPCR, they first explore the time course and concentration-dependence of downregulation of Or8 expression in Aedes in response to 1-octen-3ol, its known ligand. They then use RNASeq to identify other odorant receptors whose mRNA expression changes, as well as to investigate other transcriptional changes in response to this odorant. They provide some structural modeling data to suggest that the putative "1-octen-3-ol" receptors have binding pockets similar to known Or8 receptors.

Overall the data is clearly presented and shown. The transcriptional changes observed are likely reproducible and have strong p-values. However, the conclusions that can be drawn and the interpretation of the data needs to be reconsidered.

The DREAM technique is based on work cited by the authors from Von Der Weid et al (establishing the technique mainly in mice and for a few fly receptors) and Koerte et al (Ref 15 in manuscript). The latter work from Koerte et al investigated the reliability of the technique in flies, where receptor-ligand pairs are well characterized. Their conclusion was that "it seems highly unlikely that the application of the same experimental conditions during the DREAM treatment will work for the deorphanization of a large set of ORs, neither in D. melanogaster, nor in other insect species where novel deorphanization is necessary." Although the authors cite this study to show that some receptor-ligand pairs are correctly identified, there are nearly as many false-positives and false-negatives, and this is considering receptors that are either up- or down-regulated by the odorants.

Indeed, the data in the manuscript (S2 Table D and line 196) show that Or10 and Or4 are similarly downregulated by 1-octen-3ol exposure as the known 1-octen-3-ol receptor Or8, yet previous work on heterologously expressed Or10 and Or4 showed Or10 does not respond at all to 1-octen-3ol, and Or4 responds only weakly (Bohbot 2011 and Dekel 2019). This calls into question the utility of the DREAM approach, and the conclusion that many of the identified receptors respond to this ligand.

Other than directly demonstrating interactions between the identified receptors (probably beyond the scope of this study), the manuscript needs to be written to thoroughly consider the limitations of the DREAM approach, including their own false-positive data.

6. PLOS authors have the option to publish the peer review history of their article (what does this mean?). If published, this will include your full peer review and any attached files.

Reviewer #1: No

Reviewer #2: No

---

## [Author Response · Author response to Decision Letter 0]

10 Sep 2023

Here we provide a point by response to the editor and reviewers’ comments. Our responses are in blue text (which is included in the attached Response to Reviewers document).

Journal Requirements:

The manuscript was modified to fit the style requirements and formatting required for PLOS ONE.

This study involved funding from NIH grant, 1R21AI142140, and CDC grant, U01CK000510. No additional external funding was received for this study. Please see point 3.

"F.M., A.J.B., and M.D. were supported by the National Institute of Allergy and Infectious Disease (NIAID) of National Institutes of Health (NIH) (1R21AI142140-01). B.E. was The funders had no role in study design, data collection and analysis, decision to publish, or preparation of the manuscript."

Please respond by return e-mail so that we can amend your financial disclosure and competing interests on your behalf.

"F.M., A.J.B., and M.D. were supported by the National Institute of Allergy and Infectious Disease (NIAID) of National Institutes of Health (NIH) (1R21AI142140). B.E. was supported by The Centers for Disease Control and Prevention (CDC), Southeastern Center of Excellence in Vector-borne Disease (U01CK000510). The funders had no role in study design, data collection and analysis, decision to publish, or preparation of the manuscript. There was no additional external funding received for this study."

I have responded by email so the above financial disclosure and competing interest statement can be amended.

The RNA-seq transcriptome data is available from the NCBI Sequence Read Archive (SRA) at https://www.ncbi.nlm.nih.gov/bioproject/PRJNA942345/ and is associated with BioProject ID PRJNA942345.

5. We note that Figures 1 and 2 in your submission contain copyrighted images. All PLOS content is published under the Creative Commons Attribution License (CC BY 4.0), which means that the manuscript, images, and Supporting Information files will be freely available online, and any third party is permitted to access, download, copy, distribute, and use these materials in any way, even commercially, with proper attribution. For more information, see our copyright guidelines: http://journals.plos.org/plosone/s/licenses-and-copyright.

a. You may seek permission from the original copyright holder of Figures 1 and 2 to publish the content specifically under the CC BY 4.0 license. 

Figures 1 and 2 were replaced to ensure copyright compliance, none of the underlying data represented was modified. All images are internally generated with the exception of a modified version of https://commons.wikimedia.org/wiki/File:Aedes_aegypti_resting_position_E-A-Goeldi_1905.jpg#filelinks; which is in the public domain. 

Supporting Information file received captions and in-text citations. 

Here are additions to the references:

32.Task D, Lin CC, Vulpe A, Afify A, Ballou S, Brbic M, et al. Chemoreceptor co-expression in Drosophila melanogaster olfactory neurons. Grunwald Kadow IC, VijayRaghavan K, editors. eLife. 2022 Apr 20;11:e72599. 

33.Herre M, Goldman OV, Lu TC, Caballero-Vidal G, Qi Y, Gilbert ZN, et al. Non-canonical odor coding in the mosquito. Cell. 2022 Aug 18;185(17):3104-3123.e28. 

34. Zufall F, Domingos AI. The structure of Orco and its impact on our understanding of olfaction. Journal of General Physiology. 2018 Nov 1;150(12):1602–5. 

Reviewers' comments:

Reviewer's Responses to Questions

Comments to the Author

1. Is the manuscript technically sound, and do the data support the conclusions?

Reviewer #1: Yes

Reviewer #2: Partly

The conclusion was modified to better fit the current evidence and is addressed in the reviewer's specific comments.

2. Has the statistical analysis been performed appropriately and rigorously?

Reviewer #1: Yes

Reviewer #2: Yes

Thank you for indicating our statistical analysis has been performed appropriately and rigorously.

3. Have the authors made all data underlying the findings in their manuscript fully available?

Reviewer #1: Yes

Reviewer #2: Yes

The RNA-seq transcriptome data is available from the NCBI Sequence Read Archive (SRA) at https://www.ncbi.nlm.nih.gov/bioproject/PRJNA942345/ and is associated with BioProject ID PRJNA942345.

4. Is the manuscript presented in an intelligible fashion and written in standard English?

Reviewer #1: Yes

Reviewer #2: Yes

Thank you for your edits to ensure an intelligible manuscript. 

5. Review Comments to the Author

Reviewer #1: The authors have assessed the feasibility of using single odorant exposure and subsequent differential expression analyses of control vs. exposed olfactory tissues to identify a cognate olfactory receptor(s) in an important vector of human disease. The project is well designed and data accurately reported. This work is easily worthy of publication in PLoS One and will draw interest from many researchers in the field of chemical ecology. My main criticism is that it is uncertain whether this approach is broadly useful for determining cognate receptors. I agree that the approach appears to narrow the search, and that when paired with protein structure modelling, the set of candidate genes becomes even smaller. On the other hand, there are still many potential genes in that candidate set and it is unclear whether these results would be similar for broadly tuned ORs. It may be more appropriate to steer the conclusion towards the broad array genes with altered expression as uncovering an unappreciated feature of neurosensory tissues.

We thank the reviewer for their overall support of the manuscript. We agree with the reviewer that our enthusiasm for the utility of this approach in deorphanizing olfactory receptors needs to be more measured. We have altered the manuscript to provide more context for our results.

Part 1 of the experiment exposes females Aedes aegypti to 1% 1-octen-3-ol for varying lengths of time and concentration to optimize effect, with 6 hours chosen as an ideal length of exposure. The researchers measured OR8 expression via qRT-PCR. This optimization was important before investing in full transcriptomes. The n-values reported were appropriate. The “no observable adverse effect level” NOAEL control was a good choice. A 1% odor concentration was well chosen for this study.

We thank the reviewer for their support of our approach.

Part 2 exposes experimental females to the odorant and other females to control for 6 hours followed by RNA-seq in three biological replicates per condition. 4.5% of genes were differentially expressed. This type of broad genetic shift makes identifying cognate receptors more difficult. The functions and pathways identified in this analysis largely classified into two general groups: those involved in chemoreception and those primarily involved in xenobiotic-like responses. The rationale for focusing on these two sets was well reasoned.

We thank the reviewer for their support of our analysis.

Part 3 conducts a structural analysis of binding pocket similarity (Alphafold2 predicted protein structure) to determine if the candidate receptor list could be trimmed further to identify the true receptor-ligand pairings. These experiments seem exhaustive given current technological limitations. Heterologous expression of single receptors or sets of receptors or live calcium imaging would be laborious and time-consuming extensions of this work.

We thank the reviewer for not requesting additional experiments in this regard. We are aware that such experiments should be done in future studies.

Specific comments:

Line 342 - check wording

Sentenced rephrased to the statement below:

“For qRT-PCR experiments, samples preservation was accomplished by snap-freezing using a dry ice-ethanol bath.”

Line 182 - "Differentially expressed genes associated with chemoreception"

In the first paragraph in this section, it may be better to discuss why OR8 was not the most depleted OR. Which other ORs are known to be co-expressed with OR8 in specific sensory neurons in Aedes? Perhaps the other depleted ORs you have identified are simply the result of differential neuronal activity over the 6 hours?

Regarding Or8 not being the most depleted: the correlation coefficient between mRNA downregulation and sensitivity as measured through electrophysiology is unclear. We therefore cannot assume that Or8 would be ordinally the most responsive because it is a known strong responder. Koerte et al [Ref 15] concluded that modulation of mRNA could correctly predict receptor-ligand interaction 69%, the direct linearity of this relationship however needs further study. Furthermore, several additional factors may play a role such as expression location and co-expression of other receptors within the same neuron. 

We added two paragraphs to that address the question posed above regarding what is known about co-expression of ORs and expression location.

“Expression of the 24 ORs identified have been found to be largely restricted to the antenna, apart from Or8 which is primarily found in the maxillary palp, and Or4 whose expression was comparatively high in rostrum tissue as well as antennal tissue[12]. The overall trend of all ORs was towards a downregulation event of transcript abundance. Of note is that Orco, the obligate co-receptor of ORs was downregulation by a log2 fold-change of -1.26 and that the 73 ORs above p >.01 averaged a log2 fold-change of – 0.52. Of the ORs not differentially expressed, evaluation in Drosophila suggests that some ligand responders will not differentially express after odor exposure, this would constitute false negatives and will go unidentified when using a DREAM or OET screening approach[15].”

“Efforts to understand the expression patterns of ORs indicate a much more complicated organization pattern than the canonical one-receptor one-neuron[32,33] Exanimation of cell clusters from sensory appendages finds robust co-expression of ORs. If co-expressed ORs operate under similar regulatory mechanisms is still unclear, however amongst the differentially expressed ORs in our OET study, Or88-Or36, Or36-Or52-Or72, Or72-Or115-Or114, and Or100-Or94 were found to be co-expressed [33]. Understanding whether co-expression results in odor-gated ion channels composed of multiple tuning ORs with different ligand-specific regions could be helpful in the interpretation of our results[34].”

Differential neuronal activity over the 6 hours is unlikely to explain all the changes detected primarily because RNA-seq comparisons are between a treated and non-treated group. Although OR fluctuation over time is known, the controls used should account for this change. 

Line 284 - “Generation and evaluation of an odor-evoked transcriptome induced by 1-octen3-ol exposure in Ae. aegypti support its role as an initial screening tool for identifying potential ligand-receptor pairings.”

My suggestion would be to rephrase this final conclusion slightly. This technique may or may not be a useful initial screening tool based on the data presented here. Its usefulness is dependent on whether or not the 24 OR candidates identified here have some functional relationship with the odorant 1-octen3-ol. OR transcript depletion could be an indirect result of 1-octen3-ol-evoked activity in a subset of ORNs. Further, RNA-seq is somewhat expensive in non-model organisms. Even still, the expression pattern revealed here is fascinating and could point to biologically relevant changes occurring downstream of odorant exposure. I think the take-home message should be that long exposures to single odorants lead to complex changes in gene expression, some of which may reveal the genetic architecture of the peripheral olfactory system.

We agree that the conclusion needs to be rephrased and find the reviewer's “take-home message” to be a very insightful and succinct summarization of our key points.

“Of the 24 ORs, Or8 was 11th, Or4 was 19th, and Or10 was 20th most downregulated. Or8, Or4 and Or10 are some of the best-characterized receptors in Ae. aegypti and only Or8 responds strongly to 1-octen-3-ol while Or4 and Or10 respond to sulcatone, and indole, respectively[4,8,30]. Here, Or4 and Or10 represent false positives which suggests that OET data is not always a predictive indicator of odor-ligand receptor relationships.”

“Most investigations into olfactory receptor responsiveness are “blind deorphanizations”, that is, attempted without knowledge or prediction of potential receptor-ligand relationships. Generation and evaluation of an odor-evoked transcriptome induced by 1-octen3-ol exposure in Ae. aegypti demonstrated that prolonged exposure to one odorant could cause complex alterations in gene expression, some which may shed light on the regulatory architecture of the peripheral olfactory system. Our dataset however identified several known false positives if used as a sole indicator of ligand-receptor relationships, with the possibility of several undetermined false negatives. The limitations of this strategy as an initial screening tool to drive hypothesis-driven deorphanization warrant further investigation. Given its conservation across taxa, odor-evoked transcript responsiveness is likely an inherent property of olfaction and further study of this phenomenon may provide insight into the mechanisms underlying odor adaptation.”

We also removed this statement from the Abstract: 

“Furthermore, odor-evoked transcriptomics create a potential screening tool for filtering and identification of chemosensory and xenobiotic targets of interest”.

Even in its current form, I believe this to be worthy of publication. The paper is very well written and the data is presented transparently and without agenda. This is great work. I leave it to the editor to decide whether the main conclusion should be revisited.

Reviewer #2: The authors present work using the DREAM (Deorphanization of receptors based on expression alterations of mRNA levels) technique to identify potential receptors that respond to the ecologically important odorant 1-octen-3ol in Aedes mosquitoes. Using qPCR, they first explore the time course and concentration-dependence of downregulation of Or8 expression in Aedes in response to 1-octen-3ol, its known ligand. They then use RNASeq to identify other odorant receptors whose mRNA expression changes, as well as to investigate other transcriptional changes in response to this odorant. They provide some structural modeling data to suggest that the putative "1-octen-3-ol" receptors have binding pockets similar to known Or8 receptors.

Overall the data is clearly presented and shown. The transcriptional changes observed are likely reproducible and have strong p-values. However, the conclusions that can be drawn and the interpretation of the data needs to be reconsidered.

The DREAM technique is based on work cited by the authors from Von Der Weid et al (establishing the technique mainly in mice and for a few fly receptors) and Koerte et al (Ref 15 in manuscript). The latter work from Koerte et al investigated the reliability of the technique in flies, where receptor-ligand pairs are well characterized. Their conclusion was that "it seems highly unlikely that the application of the same experimental conditions during the DREAM treatment will work for the deorphanization of a large set of ORs, neither in D. melanogaster, nor in other insect species where novel deorphanization is necessary." Although the authors cite this study to show that some receptor-ligand pairs are correctly identified, there are nearly as many false-positives and false-negatives, and this is considering receptors that are either up- or down-regulated by the odorants.

We agree the overall conclusion should be rephrased to consider the limitation that has been shown in previous work. We overall agree with the conclusion of Koerte et al "it seems highly unlikely that the application of the same experimental conditions during the DREAM treatment will work for the deorphanization of a large set of ORs”. 

However, we do not believe the potential utility is in the attempt to deorphanize “large sets of ORs”. Their finding that modulation of mRNA could correctly predict receptor-ligand interaction 69%, suggests that a measured approach attempting to find “some” (not all) receptors using a hypothesis-driven strategy is likely improved using an OET approach. The current state of deorphanization is “blind”, meaning it occurs in the absence of any hypothesis regarding the ligand-receptor relationship. If one was to attempt to find the cognate receptors to a single behaviorally salient odor of interest, for example, they would do so simply by trial and error in an orphan olfactome of 100+ receptors potentially. Their finding suggests that the number of attempts needed to find “some” hits to this single behaviorally salient odor is likely improved first by generating a candidate list. We agree that even in the best-case scenario it is not suitable as a deorphanization for deducing all ligand-receptor relationships. 

Indeed, the data in the manuscript (S2 Table D and line 196) show that Or10 and Or4 are similarly downregulated by 1-octen-3ol exposure as the known 1-octen-3-ol receptor Or8, yet previous work on heterologously expressed Or10 and Or4 showed Or10 does not respond at all to 1-octen-3ol, and Or4 responds only weakly (Bohbot 2011 and Dekel 2019). This calls into question the utility of the DREAM approach, and the conclusion that many of the identified receptors respond to this ligand. Other than directly demonstrating interactions between the identified receptors (probably beyond the scope of this study), the manuscript needs to be written to thoroughly consider the limitations of the DREAM approach, including their own false-positive data.

The overall conclusion was rephrased to account for the limitation of indicated by the dream approach and to reflect the false positives and false negative present when interpreting the OET data. Below highlighted in blue are additions that address these concerns.

“Of the 24 ORs, Or8 was 11th, Or4 was 19th, and Or10 was 20th most downregulated. Or8, Or4 and Or10 are some of the best-characterized receptors in Ae. aegypti and only Or8 responds strongly to 1-octen-3-ol while Or4 and Or10 respond to sulcatone, and indole, respectively[4,8,30]. Here, Or4 and Or10 represent false positives which suggests that OET data is not always a predictive indicator of odor-ligand receptor relationships.”

“Most investigations into olfactory receptor responsiveness are “blind deorphanizations”, that is, attempted without knowledge or prediction of potential receptor-ligand relationships. Generation and evaluation of an odor-evoked transcriptome induced by 1-octen3-ol exposure in Ae. aegypti demonstrated that prolonged exposure to one odorant could cause complex alterations in gene expression, some which may shed light on the regulatory architecture of the peripheral olfactory system. Our dataset however identified several known false positives if used as a sole indicator of ligand-receptor relationships, with the possibility of several undetermined false negatives. The limitations of this strategy as an initial screening tool to drive hypothesis-driven deorphanization warrant further investigation. Given its conservation across taxa, odor-evoked transcript responsiveness is likely an inherent property of olfaction and further study of this phenomenon may provide insight into the mechanisms underlying odor adaptation.”

“Of the ORs not differentially expressed, evaluation in Drosophila suggests that some ligand responders will not differentially express after odor exposure, this would constitute false negatives and will go unidentified when using a DREAM or OET screening approach[15]..”

We also removed this statement from the Abstract: 

“Furthermore, odor-evoked transcriptomics create a potential screening tool for filtering and identification of chemosensory and xenobiotic targets of interest”.

---

## [Decision Letter · Decision Letter 1]

4 Oct 2023

Odor-evoked transcriptomics of Aedes aegypti mosquitoes

PONE-D-23-09227R1

Dear Dr. DeGennaro,

We’re pleased to inform you that your manuscript has been judged scientifically suitable for publication and will be formally accepted for publication once it meets all outstanding technical requirements.

Kind regards,

Efthimios M. C. Skoulakis, PhD

Academic Editor

PLOS ONE

Additional Editor Comments (optional):

Reviewers' comments:

Reviewer's Responses to Questions

**Comments to the Author**

1. If the authors have adequately addressed your comments raised in a previous round of review and you feel that this manuscript is now acceptable for publication, you may indicate that here to bypass the “Comments to the Author” section, enter your conflict of interest statement in the “Confidential to Editor” section, and submit your "Accept" recommendation.

Reviewer #2: All comments have been addressed

2. Is the manuscript technically sound, and do the data support the conclusions?

Reviewer #2: Yes

3. Has the statistical analysis been performed appropriately and rigorously? 

Reviewer #2: Yes

4. Have the authors made all data underlying the findings in their manuscript fully available?

Reviewer #2: Yes

5. Is the manuscript presented in an intelligible fashion and written in standard English?

Reviewer #2: Yes

6. Review Comments to the Author

Reviewer #2: (No Response)

7. PLOS authors have the option to publish the peer review history of their article (what does this mean?). If published, this will include your full peer review and any attached files.

Reviewer #2: No

---

## [Editor Report · Acceptance letter]

13 Oct 2023

PONE-D-23-09227R1 

Odor-evoked transcriptomics of *Aedes aegypti* mosquitoes 

Dear Dr. DeGennaro:

I'm pleased to inform you that your manuscript has been deemed suitable for publication in PLOS ONE. Congratulations! Your manuscript is now with our production department. 

Kind regards, 

on behalf of

Dr. Efthimios M. C. Skoulakis 

Academic Editor

PLOS ONE